# COVID-19: The Potential Treatment of Pulmonary Fibrosis Associated with SARS-CoV-2 Infection

**DOI:** 10.3390/jcm9061917

**Published:** 2020-06-19

**Authors:** Kacper Lechowicz, Sylwester Drożdżal, Filip Machaj, Jakub Rosik, Bartosz Szostak, Małgorzata Zegan-Barańska, Jowita Biernawska, Wojciech Dabrowski, Iwona Rotter, Katarzyna Kotfis

**Affiliations:** 1Department of Anaesthesiology, Intensive Therapy and Acute Intoxications, Pomeranian Medical University, 70-111 Szczecin, Poland; kacper.lechowicz@pum.edu.pl (K.L.); mazegan@wp.pl (M.Z.-B.); 2Department of Pharmacokinetics and Monitored Therapy, Pomeranian Medical University, 70-111 Szczecin, Poland; starkdrozd@wp.pl; 3Department of Physiology, Pomeranian Medical University, 70-111 Szczecin, Poland; machajf@gmail.com (F.M.); jakubrosikjr@gmail.com (J.R.); bartszost1@gmail.com (B.S.); 4Department of Anaesthesiology and Intensive Therapy, Pomeranian Medical University, 71-242 Szczecin, Poland; lisienko@wp.pl; 5Department of Anaesthesiology and Intensive Care, Medical University, 20-090 Lublin, Poland; w.dabrowski5@gmail.com; 6Department of Medical Rehabilitation and Clinical Physiotherapy, Pomeranian Medical University, 71-210 Szczecin, Poland; iwona.rotter@pum.edu.pl

**Keywords:** SARS-CoV-2, COVID-19, pneumonia, pulmonary fibrosis, treatment, pathophysiology, SARI, ARDS, coronavirus

## Abstract

In December 2019, a novel coronavirus, SARS-CoV-2, appeared, causing a wide range of symptoms, mainly respiratory infection. In March 2020, the World Health Organization (WHO) declared Coronavirus Disease 2019 (COVID-19) a pandemic, therefore the efforts of scientists around the world are focused on finding the right treatment and vaccine for the novel disease. COVID-19 has spread rapidly over several months, affecting patients across all age groups and geographic areas. The disease has a diverse course; patients may range from asymptomatic to those with respiratory failure, complicated by acute respiratory distress syndrome (ARDS). One possible complication of pulmonary involvement in COVID-19 is pulmonary fibrosis, which leads to chronic breathing difficulties, long-term disability and affects patients’ quality of life. There are no specific mechanisms that lead to this phenomenon in COVID-19, but some information arises from previous severe acute respiratory syndrome (SARS) or Middle East respiratory syndrome (MERS) epidemics. The aim of this narrative review is to present the possible causes and pathophysiology of pulmonary fibrosis associated with COVID-19 based on the mechanisms of the immune response, to suggest possible ways of prevention and treatment.

## 1. Introduction

In December 2019, Chinese scientists discovered a new virus, now called SARS-CoV-2, belonging to the coronavirus family and causing mainly pulmonary infections ranging from mild to severe [1]. The infection has spread very quickly, initially in the Hubei Province in China and soon all around the world. In March 2020, the Coronavirus Disease 2019 (COVID-19) was declared an epidemic by the World Health Organization (WHO) and became a global health threat [2]. According to the 101st situation report published by WHO on 30 April 2020, there were 3,090,445 confirmed infections and 217,769 deaths in the world [3]. The increase in the number of patients suffering from COVID-19 over time between 21 January 2020 and 28 April 2020 is shown in Table 1 [4].

COVID-19 spreads mainly via droplets and through direct contact [5]. For now, no other route of infection has been proven, however the fecal–oral transmission pathway has been suggested [6,7]. The conjunctival route has not been proven, but the presence of the virus has been detected in the tears [8]. To limit the spread of the disease, most governments around the world introduced quarantine as it is estimated that one person spreads the infection to an average of 2.2 to 3.58 other people in their surroundings [9]. The average incubation time of the virus is about five days, while 95% of patients will become symptomatic within 12.5 days [9,10]. Respiratory failure caused by the predilection of the SARS-CoV-2 to the respiratory tract has been defined by the WHO as severe acute respiratory infection (SARI) [11]. To date, there is no reliable data on the frequency and severity of pulmonary fibrosis associated with COVID-19.

There is currently no clear evidence of what therapy is efficacious to improve prognosis in patients infected with SARS-CoV-2. To date, numerous clinical trials are underway for drugs that have shown in vitro efficacy. Even less is known about the complications of COVID-19 infection and their treatment. The aim of this narrative review is to present the possible causes and pathophysiology of pulmonary fibrosis associated with COVID-19 based on the mechanisms of the immune response, in order to provide a review of the literature to suggest possible ways of prevention and treatment.

## 2. Clinical Features of COVID-19

Patients with SARS-CoV-2 infection range from asymptomatic carriers to those affected by a severe acute respiratory infection (SARI) [11] leading to acute respiratory distress syndrome (ARDS), which may be complicated by sepsis and death [5,9,10,12,13,14]. The range of symptoms includes fever, dry cough, myalgia, weakness and shortness of breath, but other early symptoms have been identified with time, such as gastrointestinal (i.e., watery stools, anorexia, and upper abdominal discomfort) or neurologic (i.e., delirium) [6,7,15,16]. The symptomatic course is usually associated with lower respiratory tract inflammation [5,9,12,13,14]. Pulmonary symptoms occurring in those infected can be divided into mild, moderate or severe [12]. The severe course of pneumonia in COVID-19, called SARI is characterized by shortness of breath, respiratory rate >30/min, saturation <93% and SpO2/FiO2 < 300 and warrants admission to the intensive care unit (ICU). Usually, the symptoms develop within 24 to 48 h of infection and over time can lead to life-threatening conditions [12].

In the diagnosis of SARS-CoV-2 infection, it is worth paying attention to laboratory abnormalities, which show leuko- and lymphopenia, thrombocytopenia, increased levels of lactate dehydrogenase, liver enzymes and d-dimers [17]. Procalcitonin levels are usually within normal limits [9,10,18].

Radiological imaging is extremely useful in assessing the degree of lung damage and lung parenchyma involvement and sometimes replaces laboratory tests in diagnostics. The presented images are varied, and their progression is fast [19,20,21,22]. In one of the studies, the computed tomography (CT) images of 63 patients with COVID-19 were analyzed [21]. The average number of affected pulmonary lobes was 3.3 ± 1.8, the most common changes were patchy ground-glass opacities, 17.5% patients had fibrous stripes and 12.7% patients had irregular solid nodules [23]. The vast majority of patients (85%) progressed, while the nodules and stripes enlarged [19,21,22]. Pulmonary fibrosis may be one of the major complications in COVID-19 patients [19].

Laboratory testing performed in patients with CoV infections has been performed either routinely to aid the diagnosis or to identify the progression of the disease for research purposes. Huang et al. evaluated the level of cytokines in patients with COVID-19 and compared their levels in severely ill patients in the ICU with the levels in non-ICU patients. Compared with non-ICU patients, ICU patients had higher plasma levels of IL-2, IL-7, IL-10, GSCF, IP-10, MCP1, MIP1A and TNF-α [14]. The increase in proinflammatory cytokines (e.g., IL-1B, IL-6, IL-12, IFN-γ, IP-10 and MCP1) in patients with SARS-CoV was associated with pneumonia and extensive lung damage [24].

In one scientific study, Susanna et al. investigated the cytokine response associated with MERS-CoV infection compared to SARS-CoV infection, by measuring mRNA expression levels of eight cytokine genes. The research was conducted using cells of the Calu-3 line (polarized airway epithelium Calu-3) infected with MERS-CoV and SARS-CoV at 4, 12, 24 and 30 h. Out of eight cytokines tested, six (IL-1b, IL-6, IL-8, TNF-α, IFN-β and IP-10) showed significantly increased expression in MERS-CoV and/or SARS-CoV infected Calu-3 cells when compared to uninfected cells. Among these six cytokines, proinflammatory cytokines, IL-1b, IL-6 and IL-8, induced by MERS-CoV showed significantly higher expression than those induced by SARS-CoV after 30 h. However, the levels of TNF-α, IFN-β and IP-10, which are important for the innate antiviral immune response, were significantly higher in cells induced by SARS-CoV than those induced by MERS-CoV after 24 and 30 h. The other two cytokines, MCP-1 (chemokine) and TGF-β (anti-inflammatory cytokine), showed no obvious increase after MERS-CoV or SARS-CoV infection [25].

A summary provided in Table 2 shows cytokine levels in COVID-19 and Severe Acute Respiratory Syndrome (SARS) or Middle East Respiratory Syndrome (MERS) studies. This similarity suggests that the course of SARI during COVID-19 infection is analogous to SARS or MERS, so we can suggest proceedings based on previous experiences.

## 3. SARS/MERS Outbreaks and Pulmonary Fibrosis 

Given the threat of outbreaks of severe coronavirus disease, it is important to analyze pulmonary complications associated with the previous epidemics of SARS and MERS that preceded the current COVID-19. The aforementioned outbreaks were associated with tragic results: SARS ended with 8098 cases and 774 deaths [26] and MERS—with 2206 cases and more than 787 deaths [27]. SARS and MERS treatment involved only supportive care due to a lack of effective specific antiviral treatments or vaccines [26,27]. Clarification of mechanisms responsible for pulmonary fibrosis during COVID-19, currently not well understood and not adequately treated, might improve therapeutic countermeasures [28]. For research purposes, constructing a reproducible animal model might be crucial in diminishing the consequences of emerging coronaviral diseases [29].

SARS infection spreads mainly through inhalation of respiratory droplets and rarely via fecal–oral transmission [26]. Apart from femoral head necrosis, pulmonary fibrosis, which occurs during lung healing [28], is the most severe distant consequence of SARS [30]. During SARS, different morphologic lesions were observed in the lungs. They may be subcategorized into three phases: (1) acute exudative inflammation (focal fibroplasia and reticulin fiber formation already present); (2) fibrous proliferation (similar to proliferative interstitial pneumonia; mesenchymal cells differentiate into myofibroblasts and fibroblasts) and (3) final fibrotic stage with numerous types I and IV collagen fibers [31]. The primitive mesenchymal cells, hyperplastic alveolar epithelial cells, the epidermal growth factor receptor (EGFR) and macrophages are crucial in the pathogenesis of fibrosis during SARS [28,31,32]. Patients first develop atypical pneumonia, followed by acute lung injury (ALI) and acute respiratory distress syndrome (ARDS), which evolves into fibrosis [33]. Fibrosis occurs more often amongst the older population [28,32] and patients with a severe course of the disease [34]. Fibrosis is correlated with disease duration [35]; however, it may resolve spontaneously [36,37,38]. 

According to Zhang et al., who reported a 15-year follow-up of 80 SARS patients, out of whom 71 completed the follow-up, pulmonary lesions diminish only during the first year after infection (*p* < 0.001). At the beginning of the study 9.4% (1.57%–17.23%), after one year 3.2%  ± 4.78% and after 15 years 4.60% ± 6.37% of patients had pulmonary lesions visible on CT scans [30]. As reported by Zhang et al. pulmonary interstitial damage caused by SARS mostly recovered [30].

Similar findings were reported for MERS by Das et al. [39]. When analyzing data regarding 36 patients diagnosed with SARS, the follow-up at 32 to 230 days (median 43 days) showed that lung fibrosis developed in a substantial number of convalescents, and that older patients in severe condition hospitalized in the ICU are at greater risk of being diagnosed with this complication [39]. Nevertheless, some differences between lesions in MERS and SARS have been reported [25,40].

In one scientific study, Lau et al. investigated the cytokine response associated with MERS-CoV infection compared to SARS-CoV infection, by measuring mRNA expression levels of eight cytokine genes. The research was conducted using cells of the Calu-3 line (polarized airway epithelium Calu-3) infected with MERS-CoV and SARS-CoV at 4, 12, 24 and 30 h. Out of eight cytokines tested, six (IL-1b, IL-6, IL-8, TNF-α, IFN-β and IP-10) showed significantly increased expression in MERS-CoV and/or SARS-CoV infected Calu-3 cells when compared to uninfected cells. Among these six cytokines, proinflammatory cytokines, IL-1b, IL-6 and IL-8, induced by MERS-CoV showed significantly higher expression than those induced by SARS-CoV after 30 h. However, the levels of TNF-α, IFN-β and IP-10, which are important for the innate antiviral immune response, were significantly higher in cells induced by SARS-CoV than those induced by MERS-CoV after 24 and 30 h. The other two cytokines, MCP-1 (chemokine) and TGF-β (anti-inflammatory cytokine), showed no obvious increase after MERS-CoV or SARS-CoV infection [25].

The degree of pulmonary fibrosis is positively correlated with the duration of SARS-CoV-1 disease [35]. Clinical data have shown that fibrous organization is more common in patients at the late stage than in patients at an early or mid-term stage. Importantly, pulmonary fibrosis was even seen in patients with SARS who had recovered and were discharged from the hospital. In addition, the incidence of pulmonary fibrosis was approximately 21.5% (67/311) in SARS patients who recovered after nine months from discharge from the hospital [37,38].

## 4. COVID-19 and Pulmonary Fibrosis

It has been reported that SARS-CoV-2 uses angiotensin-2-converting enzyme (ACE2) as a cell receptor in humans, causing interstitial lung damage at first and then parenchymal lesions [41]. There is a hypothesis based on the results of an experiment on the Vero-E6 cell line that supplying the soluble form of ACE2 may be associated with reduced viral infection [42,43]. It has been suggested that pulmonary complications of coronaviruses infection could be inhibited at an early stage [21]. A similar effect might be achieved through pharmacological interference of TMPRSS2 host protein [44,45,46]. Consistently, studies on the tissue distribution of ACE2 suggest that the virus receptor is widely expressed in human tissue including the digestive tract, kidney, testis and other organs [47].

Pulmonary fibrosis is a pathological consequence of acute and chronic interstitial lung diseases. It is characterized by unsuccessful reconstruction of the damaged alveolar epithelium, persistence of fibroblasts and excessive deposition of collagen and other extracellular matrix (ECM) components (e.g., ECM), as well as the destruction of normal lung architecture [48]. The progression of pulmonary fibrosis results in a widening of the interstitial matrix, final compression and destruction of normal pulmonary parenchyma, and thus damage to capillaries leading to respiratory failure [49]. The etiology of pulmonary fibrosis is multifactorial and includes age, smoking, viral infection, drug exposure and genetic predisposition [35,50]. An additional mechanism may be oxidative stress associated with excessive reactive oxygen species (ROS) production. This may be due to improper removal of ROS (aging) or associated with an excessive supply of a high percentage of oxygen, e.g., shortness of breath due to COVID-19 infection. An increase in apoptosis of follicular cells associated with oxidative stress has been observed in idiopathic pulmonary fibrosis (IPF). TGF-β1 contributes to the development of fibrosis and the production of ROS [51,52]. Moreover, tyrosine kinases signaling mediated by fibroblast growth factor (FGF) and platelet-derived growth factor (PDGF) is also crucial in pulmonary fibrosis development. The studies determining IPF pathogenesis reported that overexpression of FGF and PDGF increases the pulmonary fibroblast proliferation. Furthermore, FGF enhances the profibrotic effects of TGF-β1 [53]. 

Chronic inflammation has been considered as the main cause of pulmonary fibrosis and may lead to epithelial damage and fibroblast activation. Other studies suggest that alveolar epithelial damage and the formation of active myofibroblast foci are the main causes of most lung fibrosis processes [35]. Viral infections may act as triggers for the initiation of IPF or as agents exacerbating existing fibrosis. Especially the elderly population is prone to viral-induced fibrosis due to immunosenescence and with viral infections acting as cofactors [50]. 

Pulmonary fibrosis presents with the following symptoms: dry cough, fatigue and dyspnea. Patients might lose weight and their physical condition deteriorates. Therefore, people suffering from this disorder might lose their source of income and their quality of life systematically regresses. Treatment of COVID-19 and other coronaviral diseases cannot exclude prophylaxis or therapy of pulmonary fibrosis in order to provide a satisfactory distant prognosis.

When damage occurs in the lung tissue, a set of growth factors and cytokines, including monocyte-1 chemoattractant protein (MCP-1), transforming growth factor β1 (TGF-β1), tumor necrosis factor a (TNF-α), fibroblast growth factor (FGF), platelet-derived growth factor (PDGF), interleukin-1b (IL-1b) and interleukin-6 (IL-6), are overexpressed and released by cells [49,53]. The recent reports show that serum levels of the aforementioned cytokines and growth factors are also highly increased in COVID-19 patients [54,55,56]. 

Dysregulated release of matrix metalloproteinases, which causes epithelial and endothelial injury [57] and uncontrolled fibroproliferation are amongst the most important mediators of the inflammatory phase of ARDS [58]. TGF-β regulates fibrosis [59] and together with VEGF, Il-6, TNF-α and vascular dysfunction participate in progression to fibrosis [58,60,61]. This process does not occur amongst all patients [2]. It is possible that increased levels of proinflammatory cytokines amongst older people are responsible for a more severe course of the disease in this group of patients. The similar cytokine profiles in IPF and COVID-19 suggest analogous pathomechanisms of pulmonary fibrosis in those diseases, therefore, drugs useful in the treatment of IPF could be also beneficial for COVID-19 patients. 

Type II vesicular endothelial cells are one of the main sources of these fibrogenic factors. These factors stimulate hyperproliferation of type II follicular cells, recruit fibroblasts to fibrotic loci, and induce differentiation and activation of fibroblasts into myofibroblasts. Myofibroblasts are responsible for excessive ECM accumulation in basement membranes and interstitial tissues, which ultimately leads to loss of alveolar function, especially the gas exchange between alveoli and capillaries [49]. The mechanism of fibrosis formation is presented in Figure 1.

## 5. Potential Treatment Options in COVID-19 Patients

While ARDS seems to be the main cause of pulmonary fibrosis in COVID-19, several reported have mentioned that ARDS caused by SARS-CoV-2 is different from the typical ARDS [62]. It seems that COVID-19 induced ARDS differs substantially from that caused by other factors as high compliance is often observed, which is inconsistent with the severity of hypoxia. Several reports suggest that the onset of COVID-19 ARDS was between 8 and 12 days, rendering the one-week onset limit inadequate [13,63]. Moreover, CT findings in many cases are not suggestive of “typical” ARDS [64]. In addition, abnormal coagulopathy is apparent—a procoagulant pattern is present in numerous patients [65]. It is, therefore, suspected that the mechanism of pulmonary fibrosis in COVID-19 is different from that of IPF and other fibrotic lung diseases, especially with pathological findings pointing to alveolar epithelial cells being the site of injury, and not the endothelial cells [63,66]. These facts indicate that other specific therapeutic options should be considered in COVID-19.

There is currently no fully documented therapeutic method in the treatment of postinflammatory pulmonary fibrosis after coronavirus infection. However, some therapies may be considered.

### 5.1. Steroids

The rationale for the use of corticosteroids in viral pneumonia is to decrease the host inflammatory response in the lungs, as this may lead to the development of acute lung injury and ARDS. Direct evidence for the use of corticosteroids in COVID-19 is very limited, however previous reviews regarding outcomes in other viral pneumonia provide important clinical information [67,68]. Studies regarding SARS and MERS outbreaks reported delayed viral clearance from blood and respiratory tract [67,69]. The authors reported two important clinical observations—no influence of corticosteroids on survival improvement with concomitant steroid-related adverse events, such as avascular necrosis causing femoral head necrosis, hyperglycemia typically occurring during steroid treatment and psychosis [67,69]. A meta-analysis including patients with influenza pneumonia reported that the use of corticosteroids was associated with an increased risk of mortality and secondary infections [70]. Wu et al. reported that in patients with COVID-19 who developed ARDS, treatment with methylprednisolone was associated with a decreased risk of death [71]. Current research suggests not to administer systemic steroids during CoV infections, as they might delay clearance of viral RNA and cause an increased risk of secondary infection [69,72], if there are no other reasons for this type of treatment, including refractory shock or chronic obstructive pulmonary disease [68,73]. Such a specific therapy will modify the course of this condition if infection plays a crucial role in the fibrosis development.

### 5.2. Spironolactone

There are several reports which state that the use of spironolactone may be of significant importance in fibrosis prevention [74,75,76]. Activation of the mineralocorticoid receptor (MR) is a factor contributing to the pathophysiology of many diseases. Aldosterone, which is an adrenocortical hormone and physiological MR activator, is partly responsible for the increase in extracellular matrix turnover, which is observed in pulmonary, cardiac and renal fibrosis and exerts its effect primarily on lung epithelium [74]. It is known that an increased level of aldosterone can induce hypertension, alter inflammation and fibrosis, and exacerbate cardiovascular disease [75]. The limitations of some of these studies are related to the fact that they were carried out on animal models such as rats or other rodents. There are also no direct studies that present the beneficial effects of the mineralocorticoid receptor antagonist in postviral fibrosis, but it could serve as a potential treatment for such a serious lower respiratory tract complication.

In various animal models, spironolactone has been shown to act as an antioxidant and to protect organs from damage associated with oxidative stress by strengthening the antioxidative defense systems while inhibiting free radicals production [77]. Lung tissue treated with spironolactone showed a reduced number of cells such as lymphocytes, neutrophils, macrophages and eosinophils in the alveoli compared to those in which spironolactone was not used. Lieber et al. showed that spironolactone treatment alleviates acute pneumonia caused not only by bleomycin but also by lipopolysaccharides [77]. In one preclinical study, Barut et al. analyzed the effect of spironolactone on lung damage due to intestinal ischemia and reperfusion. The results suggest that initial treatment with spironolactone reduced neutrophil infiltration, nitric oxide synthase induction, oxidative stress and histopathological damage. Similarly, Atalay et al. demonstrated the effectiveness of spironolactone in the treatment of acute lung damage [78], while Ji et al. indicated the therapeutic potential of spironolactone, which significantly reduces the inflammatory response of the lungs caused by bleomycin [79,80].

### 5.3. Fibrinolytic Agents

ARDS is the most common pulmonary complication of the COVID-19. There is no effective treatment other than supportive therapy [81]. A new discovery in the pathophysiology of ARDS is the fibrin deposition in the air spaces and lung parenchyma, together with platelet thrombotic microclots in the pulmonary vessels, which contribute to the development of progressive respiratory failure and right heart failure [82]. Similar pathological results were currently observed in lung samples from patients infected with COVID-19 [83]. This destructive activation of the coagulation system in ARDS results from increased activation and mobilization of clot formation together with inhibition of fibrinolysis processes and is thought to mediate pulmonary endothelial dysfunction in the case of influenza A [84]. Since the year 2003, it has been proposed to target the coagulation and fibrinolysis systems to improve ARDS treatment [85]. In particular, the use of plasminogen activators to reduce the progression of ARDS and to reduce death induced by this pathology has provided promising evidence from animal models and a phase 1 clinical study in humans [86,87]. In the year 2001, Hardaway and colleagues showed that urokinase or streptokinase administration in patients with terminal ARDS reduced the expected mortality from 100% to 70% without any undesirable bleeding. Importantly, the majority of patients who eventually died, secondary to multiorgan failure, including renal or hepatic insufficiency rather than pulmonary insufficiency [87].

The inclusion of therapies that are widely available but not included as indications in the summary of product characteristics and traditionally considered “high-risk therapies” such as fibrinolytic agents is justified in this unprecedented public health threat associated with SARS-CoV-2 infection. A more modern approach to thrombolytic therapy involves the use of tissue plasminogen activator (tPA) due to the higher efficiency of clot lysis with a comparable risk of bleeding with other fibrinolytic factors. In addition, tPA treatment was found to have lower mortality, a greater increase in arterial pO2, and a greater decrease in arterial pCO2 compared to untreated controls than urokinase plasminogen activator (uPA) or plasmin in a comprehensive meta-analysis of animal studies. These studies concerned acute lung injury, but none of these studies included virus-induced ARDS [88].

The dose, route of administration and duration of treatment remain to be determined, and studies in this direction are ongoing [89]. In animal models of acute lung injury, intratracheal and intravenous administration of fibrinolytic agents was more effective than administration by aerosol. Based on extensive experience in the use of tPA in acute conditions such as stroke or heart attack, intravenous administration may be the easiest to implement [90]. In the initial state of ARDS, the authors of one publication suggest administration of 25 mg tPA within 2 h, followed by an infusion of 25 mg tPA another 22 h, at a dose not exceeding 0.9 mg/kg. The same exclusion criteria currently used to treat stroke and myocardial infarction can be used, and responders receive heparin infusion for some time after stopping treatment with tPA [89]. Patients with COVID-19 induced ARDS who have a PO_2_/FiO_2_ ratio <50 and pCO2 >60 mmHg, despite prone positioning and optimal mechanical ventilation support, appear to be candidates for tPA treatment, especially in environments where ECMO is not widely available. In addition, in scenarios where no further mechanical ventilation is possible, this may be appropriate as rescue therapy for people with progressive deterioration of lung function [89]. It is important to remember that thrombocytopenia may occur in patients with COVID-19 [18], which reduces the possibility of using this therapy.

### 5.4. Antiviral Agents

In the prevention of fibrosis, it is important to reduce the viral load and hence the duration of viral pneumonia. Development of efficacious therapeutic strategies requires the pursuit of at least one of the three concepts: clinical tests of currently known antiviral agents [91,92], usage of molecular libraries and databases [92,93] and target therapy to disrupt the functioning of the virus [92,94]. These methods led to research on therapeutic agents, out of which hydroxychloroquine and remdesivir seem to be the most promising ones.

Chloroquine and hydroxychloroquine disrupt coronavirus cell receptor glycosylation and modulate the immune system [95]. The latter proved efficient in vitro against SARS-CoV [96]. Another experience comes from the treatment of multidrug-resistant Gram-positive bacteria, the use of teicoplanin inhibits the first stages of the coronaviruses cycle in human cells [97]. Currently known antiviral agents like remdesivir, lopinavir/ritonavir might inhibit RNA replication due to high affinity to viral enzymes, reverse transcription or protein biosynthesis through premature termination or inhibition of nitrogenous bases synthesis [92,98]. Remdesivir was previously efficacious in restricting lung injury, improving medical conditions and inhibiting viral replication in the animal model of MERS infection [99,100,101]. Grein et al. reported on the results of a clinical trial with remdesivir, which began on 25 January 2020 and ended on 7 March [102]. Remdesivir was given to patients with confirmed SARS-CoV-2 infection and oxygen saturation ≤94%. During the median follow-up of 18 days, 36 patients (68%) improved their oxygen maintenance class. Moreover, 17 out of 30 patients (57%) assisted by mechanical ventilation were extubated. A total of 25 patients (47%) were discharged and seven patients (13%) died. Mortality was 18% (6 out of 34) among patients receiving invasive ventilation and 5% (1 out of 19) among patients not receiving invasive ventilation. The risk of death was greater in patients aged 70 years or older (risk ratio compared to patients under 70 years old, 11.34; 95% CI, 1.36 to 94.17) and among patients with higher serum creatinine at baseline (risk ratio per milligrams per deciliter, 1.91; 95% CI, 1.22 to 2.99) [102]. It is safe for humans and was introduced for clinical trials [92,103]. Lopinavir/ritonavir might be effective against coronaviruses if combined with interferon-1β or ribavirin [98,104,105,106]. Umifenovir potentially prevents virus binding to the cell membrane [92,107,108].

Favipiravir (FPV) may be one of the possible treatment options for COVID-19 [109]. It has been shown that FPV is able to inhibit virus reproduction in vitro. Preliminary studies comparing favipiravir treatment with lopinavir/ritonavir reported shorter viremic clearance (4 vs. 11 days) and improved radiological picture in the FPV group [110]. Irie et al. investigated the pharmacokinetics of FPV in the context of COVID-19 treatment showed that the drug concentration was lower in critically ill patients compared to healthy individuals, indicating the need for using higher doses [111]. Lack of more accurate data requires in-depth research in this area.

The antiparasitic agent ivermectin may be another therapeutic approach worth exploring further. In an in vitro study, it showed a 99.98% reduction in viral load after 48 h of treatment, in three samples [112]. The drug is not toxic at a standard dose, and is safe for pregnant women, which makes it a strong candidate for evaluation in clinical trials [112].

Tocilizumab, a humanized monoclonal antibody against interleukin-6, is an immunosuppressive drug intended primarily for the treatment of rheumatoid arthritis [113]. In China, it was expected to have a beneficial effect on coronavirus patients with severe lung damage and elevated interleukin six levels [113,114]. In one of nonrandomized, open-label clinical studies involving 21 patients with severe or critical COVID-19 treated intravenously with tocilizumab it has been shown that in 15 out of 20 patients (75%), there was a statistically significant decrease in oxygen demand from the fifth day after receiving tocilizumab. Additionally, in 19 patients (90.5%), CT scan showed resolution of radiological abnormalities [115].

Plasma therapy in patients with viral infections, including SARS-CoV-2, is also being carried out in many countries [116,117]. It was used to treat patients with SARS-CoV-1 [118]. For example, in one of the early studies within patients with SARS (*n* = 50) had a significantly higher discharge rate by day 22 following the onset of illness (73.4% vs. 19.0%; *p* < 0.001) and lower case-fatality rate (0% vs. 23.8%; *p* = 0.049) in the convalescent plasma treatment group (*n* = 19 patients) when compared with steroid treatment group (*n* = 21) [119]. One more recent article describes five cases of critically ill patients with COVID-19 and ARDS. Following plasma transfusion, body temperature normalized within three days in four out of five patients, the SOFA score decreased, and PaO2/FiO2 increased within 12 days. Viral loads also decreased and became negative within 12 days after the transfusion [120]. One of the early studies related to the use of steroids not yet peer-reviewed data, suggest ciclesonide blocks SARS-CoV-2 ribonucleic acid replication in vitro and inhibits SARS-CoV-2 cytopathic activity. This may be of great relevance to reducing the risk of developing of COVID-19 in response to SARS-CoV-2 infection or reducing the severity of the disease. The effective concentration of ciclesonide to block SARS-CoV-2 (the cause of COVID-19) replication (EC90) was 6.3 μM [121,122].

The viral life cycle with drug attachment sites is shown in Figure 2.

### 5.5. Potential Novel Strategies

It is highly probable that one of the aforementioned therapeutic agents will become useful in coronavirus infection therapy. According to research reports, another less common approach to improve the activity of the immune system also seems promising. Colony-stimulating factors like GM-CSF might exacerbate reaction to the infection [123] and accelerate the elimination of viruses [124]. Nevertheless, it is important to remember that this approach does not seem to be appropriate in the patient in critical condition. Its usefulness in the first day of therapy of nonsevere patients is still hard to predict due to a lack of clinical data verifying the effectiveness of improving viral clearance and treatment outcomes [124]. 

Idiopathic pulmonary fibrosis differs from COVID-19-related pulmonary fibrosis however, it is worth searching for a solution to this problem within these drug groups or therapeutic options. The currently ongoing clinical trials evaluating a wide range of potentially beneficial strategies for the prevention and treatment of pulmonary fibrosis associated with COVID-19 have been summarized in Table 3.

The researchers consider not only the use of well-known agents like approved by Food and Drug Administration (FDA) for idiopathic pulmonary fibrosis therapy, like nintedanib, a tyrosine kinase inhibitor [125] or pirfenidone with a uncertain mechanism of action [126], but also other therapeutic options. On 24 March 2020, OncoArendi Therapeutics announced that it is researching a drug that can help treat pulmonary fibrosis in patients who survive COVID-19, namely OATD-01 which inhibits chitotriosidase 1 (CHIT1). This action may exhibit anti-inflammatory activity and delay the development of pulmonary fibrosis [127,128]. Studies are currently underway to determine if patients who died of COVID-19 had increased CHIT1 expression in their lung tissue, which could result in a positive effect of the drug on developing pulmonary fibrosis in this disease [127,128].

Another agent, tetrandrine, is an alkaloid with a multidirectional mechanism of action, affecting reactive oxygen species, calcium channels and caspase pathways that have been found effective in the treatment of inflammation or lung cancer [129,130]. The antifibrotic properties of Fuzheng Huayu formula, that contains six Chinese herbs, and Anluohuaxian were established in liver fibrosis treatment and are currently under trial [131,132]. 

Recent trials evaluate other treatment options like administration of mesenchymal stem cells or a human purified amniotic fluid, both previously known for anti-inflammatory, antifibrotic and regenerative abilities [133,134,135]. 

Another clinical trial aiming to test with the use of hyperbaric oxygen was estimated to start on 25 April 2020. The anti-inflammatory effects, which include decreased expression of IL-1β, IL-6 and TNF-α [136,137], could be beneficial to mitigate ARDS associated with COVID-19 and fibrosis development.

Recently, it has been suggested that COVID-19 induces hypoxic pulmonary vasoconstriction similar to high altitude pulmonary edema (HAPE), therefore medications found to be effective in HAPE, including the use of acetazolamide, nifedipine and phosphodiesterase inhibitors to decrease pulmonary vasoconstriction has been suggested [138]. It must be underlined that as new data regarding COVID-19 is emerging this approach remains controversial [139].

Jing-Yu Chen et al. proposed a new treatment for fibrosis in patients with previous COVID-19 diagnosis. Three patients underwent lung transplantation after previous confirmation of negative viremia. Two patients survived and started participating in a rehabilitation program. This is an unconventional approach to this condition however, it is worth considering in patients with a critical stage of pulmonary fibrosis in the course of COVID-19 [140].

The most important types of fibrosis treatment are shown in Figure 3.

## 6. Conclusions

COVID-19 presenting with SARI is an extremely serious disease. Rapid development and a significant, constantly growing number of patients are forcing scientists to seek new treatment options. This narrative review demonstrates similarity in pulmonary symptoms and the mechanisms of their formation, with previous forms of the coronavirus (SARS, MERS). For this reason, we should rely on the knowledge acquired during the previous epidemics. One of the complications of the respiratory system infection is pulmonary fibrosis, leading to permanent disability. There are few options available for its treatment. The most important factor in limiting pulmonary fibrosis is timely antiviral treatment and elimination of the causative agent, which is currently not possible in the absence of proper treatment. Research into attempts to limit the development of fibrosis is scarce. Trials with spironolactone have been carried out on animal models with positive results. An analysis of the mechanisms of action shows that the effect of this drug on fibrosis can be positive and it is worth considering its use in high-risk patients. The second relatively promising therapy may be the use of tPA in patients without contraindications, which reduced respiratory mortality. So far, despite years of trial, no answer has been given to the question of appropriate pulmonary fibrosis therapy. The presented treatment methods are promising however, they require a closer examination in prospective randomized trials.

## Figures and Tables

**Figure 1 jcm-09-01917-f001:**
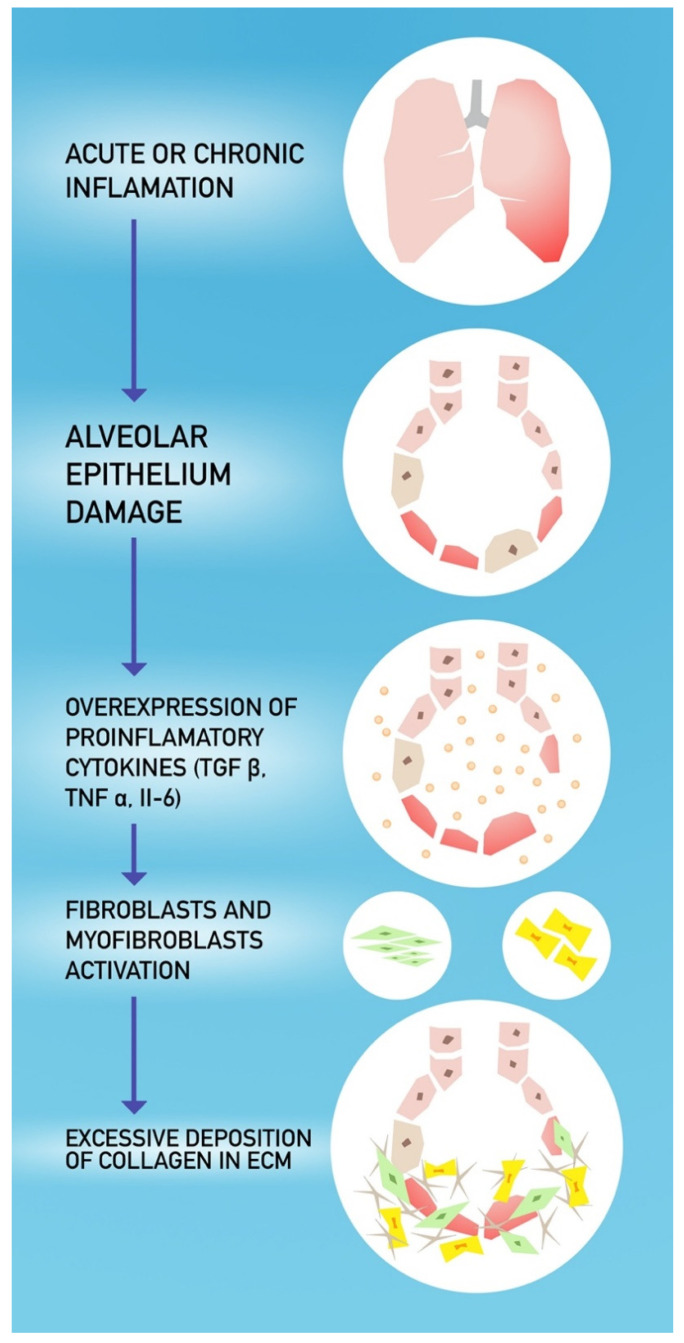
The pathogenesis of pulmonary fibrosis.

**Figure 2 jcm-09-01917-f002:**
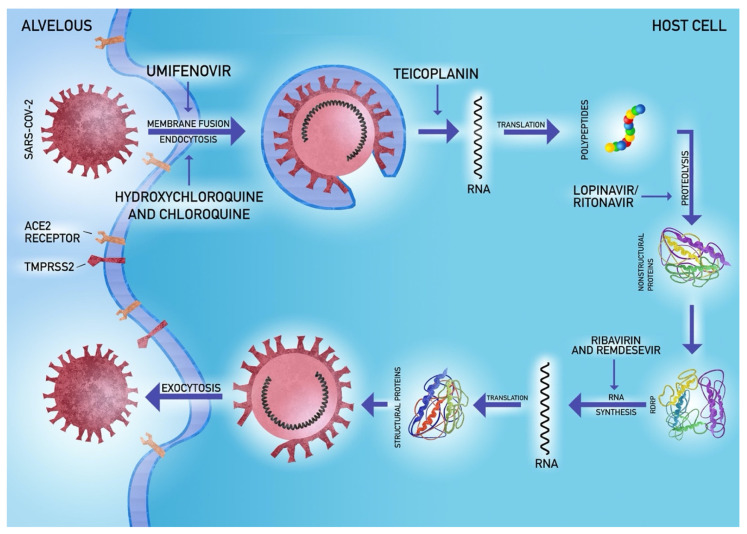
Antiviral treatment in COVID-19.

**Figure 3 jcm-09-01917-f003:**
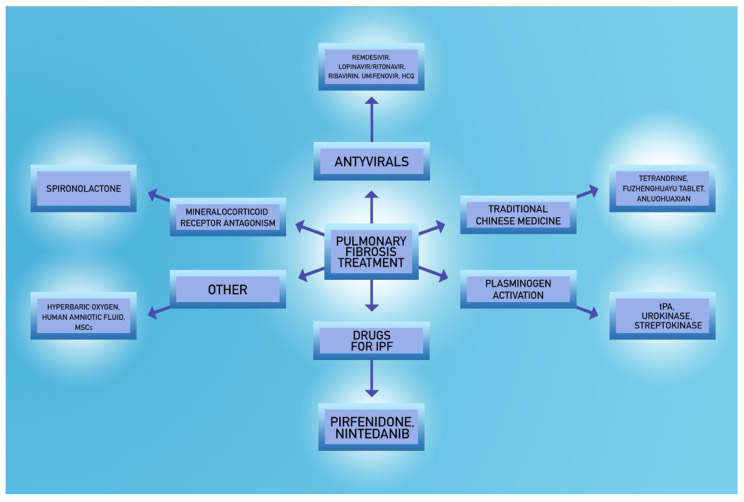
Suggested treatment options for pulmonary fibrosis in COVID-19.

**Table 1 jcm-09-01917-t001:** Weekly WHO reports regarding an increase in the number of patients suffering from Coronavirus Disease 2019 (COVID-19).

Date	Infected Patients	Number of Deaths	Mortality (%)
21 January 2020	282	6	2.13%
28 January 2020	4593	106	2.31%
4 February 2020	20,630	426	2.06%
11 February 2020	43,103	1018	2.36%
18 February 2020	73,332	1873	2.55%
25 February 2020	80,239	2700	3.36%
3 March 2020	90,869	3112	3.42%
10 March 2020	113,702	4012	3.53%
17 March 2020	179,111	7426	4.15%
24 March 2020	372,755	16,231	4.35%
31 March 2020	750,890	36,405	4.85%
7 April 2020	1,279,722	72,614	5.67%
14 April 2020	1,844,863	117,021	6.34%
21 April 2020	2,397,217	162,956	6.80%
28 April 2020	2,954,222	202,597	6.86%

**Table 2 jcm-09-01917-t002:** Cytokines overexpression in fibrosis levels in COVID-19 and severe acute respiratory syndrome (SARS)/Middle East respiratory syndrome (MERS) studies.

	Cytokines Overexpression in Fibrosis	Higher Cytokine Level in SARS/MERS	Higher Cytokine Level in COVID-19
IL-1B	+	+	+
IL-2			+
IL-6	+	+	+
IL-7			+
IL-8		+	
IL-10			+
IL-12		+	
IP-10		+	+
IFNβ		+	
IFNγ		+	
GSCF			+
MCP1	+	+	+
MIP1A			+
TNFα	+	+	+
TGFβ	+	+	+

**Table 3 jcm-09-01917-t003:** Recent clinical trials evaluating therapy of COVID-19-associated pulmonary fibrosis.

Therapeutic Agents	Clinical Trial ID	Phase	Number of Participants	Comments
Nintedanib	NCT04338802	II	96	Compared to placebo
Pirfenidone	NCT04282902	III	294	Compared to standard treatment
Tetrandrine	NCT04308317	IV	60	Compared to standard treatment
FuzhengHuayu Tablet	NCT04279197	II	136	Combined with N-acetylcysteine; compared to placebo
Anluohuaxian	NCT04334265	-	750	Compared to standard treatment
Human Amniotic Fluid	NCT04319731	I	10	-
Mesenchymal Stem Cells	NCT04288102	II	90	Compared to placebo
Hyperbaric oxygen	NCT04327505	II	200	Compared to standard treatment

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
