# Peer review of "COVID-19: The Potential Treatment of Pulmonary Fibrosis Associated with SARS-CoV-2 Infection"

_jcm, 2020, doi:10.3390/jcm9061917_

Round 1

Reviewer 1 Report

Lechowicz and colleagues presented the possible causes and pathophysiology of pulmonary fibrosis associated with COVID-19 based on the mechanisms of the  immune response, in order to provide a review of the literature to suggest possible ways of prevention and treatment. The review is clear, updated and interesting. I would suggest only to shorten a bit the “general part” to mainly focus on causes and pathophysiology of pulmonary fibrosis associated with COVID-19.

Reviewer 2 Report

1) General comments

Lechowicz et al. have described the review article entitled “COVID-19: The potential treatment of pulmonary fibrosis associated with SARS-CoV-2 infection”. SARS-CoV-2 infection is reported to cause pulmonary fibrosis, and the treatment to inhibit pulmonary fibrosis must be important in COVID-19 patients. The authors tried to clarify the mechanisms of pulmonary fibrosis in COVID-19, and mentioned some possible therapeutic options. However, there are some critical problems that need to be addressed.

2) Specific comments

Major;

The authors described the mechanisms pf pulmonary fibrosis in this manuscript. However, the content is not specific for COVID-19, and there is nothing new. ARDS is one of the causes of pulmonary fibrosis in COVID-19. Although it is sure that SARS-CoV-2 induced ARDS, several reported mentioned that ARDS caused by SARS-CoV-2 is different from the typical ARDS. For example, high compliance is often observed in ARDS caused by SARS-CoV-2. In addition, abnormal coagulopathy is apparent. In this regard, the mechanism of pulmonary fibrosis in COVID-19 should be different from that of IPF and other fibrotic lung diseases. These facts indicate that other specific therapeutic options should be considered in COVID-19. The authors must described the content for COVID-19.

Minor;

 There are lots of typos. The authors must correct them.

Reviewer 3 Report

The review article entitled “COVID-19: The potential treatment of pulmonary fibrosis associated with SARS-CoV-2 infection” describes the epidemiology, clinical features, pulmonary fibrosis associated with coronavirus diseases including SARS/MERS and COVID-19, and potential treatment options for COVID-19.

Though the contents are well written, I felt some concerns whether they are suitable for a review article which needs comprehensiveness in its contents. I think the comments attached below would help the manuscript improve in its quality.

Major;

1) The manuscript does not reflect the title of “pulmonary fibrosis” which should be a main theme in this article. This theme is described in section 3 and 4 referring to the fibrosis brought by SARS/MERS and COVID-19, and in the limited space of “potential novel strategies” in section 5. The contents and proportion of “pulmonary fibrosis” section should be enriched and widened, if this review article try to deal with “pulmonary fibrosis associated with SARS-CoV-2 infection”.

The mechanism written in section 4 should be linked with the treatment part in section 5 (Potential novel strategies). Since PDGF and FGF-2 play important roles in pulmonary fibrosis, the author should mention these growth factors in addition to TGF-β, and try to show the relationship between the pathogenesis / mechanism of COVID-19 and the fibrotic mechanism of IPF, which is important to understand why clinical trials dealing with nintedanib, pirfenidone, etc. are conducted.

2) The manuscript does not fully cover potential treatment options, especially “antiviral agents” part.

Most of the readers are interested in how to treat COVID-19 and pulmonary fibrosis following COVID-19 pneumonia. Thus, the manuscript should strengthen how physicians should treat the disease. Since the treatment of COVID-19 includes the management of pulmonary fibrosis through mitigating the lung injury caused by COVID-19, it is acceptable to mention COVID-19 treatment strategies.

In this respect, the potential treatment options should include remdesivir, favipiravir, hydroxychloroquine, ciclesonide (inhaled corticosteroid), tocilizumab, convalescent plasma, ivermectin, and nafamostat. A trial of lopinavir-ritonavir could be dealt with. The authors only mentioned chloroquine and remdesivir. The authors should fully cover these treatment options, or omit them and focus solely on the pulmonary fibrosis.

Minor;

1) The term “ECM” in line 198 (section 4) means extracellular matrix. The original manuscript is incorrect

2) There are some spelling errors as well as grammatical errors, which should be checked by native speakers.

Round 2

Reviewer 2 Report

None.

Author Response

Thank you.

Katarzyna Kotfis

Reviewer 3 Report

The review article entitled “COVID-19: The potential treatment of pulmonary fibrosis associated with SARS-CoV-2 infection” has been well revised, focusing on the management of pulmonary fibrosis following COVID-19 pneumonia.

The content has become in concordance with the title, and comprehensive enough as a review article.

I think the article could be accepted after minor revision.

Minor;

  • The newly added terms “TGF-a” and “TGF-b” are used as “TGF-α” and “TGF-β” in other parts. The terminology should be consistent throughout the manuscript.
  • The comment on favipiravir is lacking while other less important treatment candidates are described.

Author Response

Dear Reviewer, 

thank you for your valuable comments. We have improved the manuscript accordingly.

  • The newly added terms “TGF-a” and “TGF-b” are used as “TGF-α” and “TGF-β” in other parts. The terminology should be consistent throughout the manuscript.
  • Response: All terms have been corrected.

  • The comment on favipiravir is lacking while other less important treatment candidates are described.
  • Response: The following paragraph has been added: "Favipiravir (FPV) may be one of the possible treatment options for COVID-19[109]. It has been shown that FPV is able to inhibit virus reproduction in vitro. Preliminary studies comparing favipiravir treatment with lopinavir/ritonavir reported shorter viremic clearance (4 vs 11 days) and improved radiological picture in the FPV group[110]. Irie et al. investigated the pharmacokinetics of FPV in the context of COVID-19 treatment showed that the drug concentration was lower in critically ill patients compared to healthy individuals, indicating the need for using higher doses[111]. Lack of more accurate data requires in-depth research in this area."

With best regards

Katarzyna Kotfis